# Almost optimal geometrically local quantum LDPC codes in any dimension

Xingjian Li[1], Ting-Chun Lin[2,3], Adam Wills[3,4] & Min-Hsiu Hsieh[3] ✉

Geometrically local quantum codes, comprised of qubits and checks embedded in $\mathbb{R}^D$ with local check operators, have been a subject of significant interest. A key challenge is identifying the optimal code construction that maximizes both code dimension and distance under the geometric constraints. In this work, we introduce a construction that can transform any good quantum LDPC code into an almost optimal geometrically local quantum code. Our approach hinges on a novel yet simple procedure that extracts a two-dimensional structure from an arbitrary three-term chain complex, building a connection between geometric operations and code constructions. We expect that this procedure will find broader applications in areas such as weight reduction and the geometric realization of chain complexes.

In recent years, quantum coding theory has enjoyed significant progress both in theory and practice. Among various quantum codes, the quantum low-density parity-check (qLDPC) codes have drawn much attention. On the practical side, the qLDPC codes are favored since their stabilizers only act on a few qubits. The low-density property is experimentally friendly since quantum devices are sensitive to noise and measurement errors. On the theoretical side, there have been many constructions achieving optimal asymptotically linear dimension and distance recently[1–3]. Interestingly, these constructions have deep connections with high dimensional expanders.

However, to connect the theoretical development with practice, there are still several barriers. One of the barriers is that the constructions of optimal codes are based on expanders, thus the checks do not have a geometrically local embedding in the Euclidean space. Since we live in a 3D world, for certain applications, it is preferable that the code has a local embedding in $\mathbb{R}^3$. The embedding should arrange qubits and checks so that each component only connects with nearby neighbors, reflecting the physical constraints of real-world systems. To be practical, such an embedding also needs to avoid packing too many qubits into a small area.

Variants of geometrically local codes find many applications in practical systems, such as surface codes[4] and color codes[5]. Since many experimental platforms rely on local interactions for error correction operations, there have been many error correction experiments conducted based on these codes[6–8]. Constructing fault-tolerant

architectures based on these codes is an important research direction. It is also worth considering if there is any geometrically local construction with better parameters, in order to save more experimental resources.

If we take the geometric constraints into account, it is known that we cannot achieve linear dimension and distance simultaneously. In particular, Bravyi and Terhal[9] provided an upper bound on code distance $d \leq O(n^{\frac{D-1}{D}})$, and Bravyi, Poulin, and Terhal[10] showed an upper bound on the distance and rate tradeoff $k d^{\frac{2}{D-1}} \leq O(n)$ for geometrically local codes. At the time, the only known geometrically local codes were the high-dimensional toric codes, which saturate the upper bound in two dimensions but are far from optimal in higher dimensions.

Subsequent works sought to address these limitations by developing new families of codes with improved properties. Notably, Haah[11] and Michnicki[12] introduced novel constructions of geometrically local codes that surpass the toric code in certain aspects, demonstrating that there is still room for improvement within the constraints of locality. Despite these advancements, the gap between the known upper bounds and the best existing constructions remains significant.

Recent progress in quantum LDPC codes has led to new developments in constructing geometrically local codes. The first construction of (almost) good geometrically local codes is by Portnoy[13]. It first geometrizes the code by mapping the code into a manifold based on the work of Freedman and Hastings[14], taking the nerve of the manifold to obtain a 2D simplicial complex, subdividing it, and

[1]Department of Computer Science and Technology, Tsinghua University, Beijing, People's Republic of China. [2]Department of Physics, University of California, San Diego, CA, US. [3]Hon Hai (Foxconn) Research Institute, Taipei, Taiwan. [4]Center for Theoretical Physics, Massachusetts Institute of Technology, Cambridge, MA, US. ✉e-mail: min-hsiu.hsieh@foxconn.com

## Table 1 | Comparison of the constructions

| | Optimal parameters | Energy barrier | Arbitrary code | All dimensions |
|---|---|---|---|---|
| Portnoy | ✓[♯] | ✓[♭] | Almost[b] | ✓ |
| Williamson-Baspin | ✓ | ✓ | ✓ | Currently in 3D |
| Our result | ✓[♯] | ✓ | ✓ | ✓ |

[♯] Up to a polylog factor.
[♭] Not stated, but it holds.
[b] Requires a sparse $\mathbb{Z}$-lift[14,Definition 1.2.2].

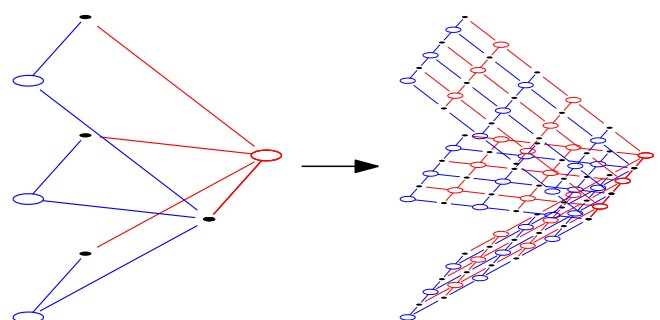

**Fig. 1 | Constructing new code by subdivision.** The blue, black, red vertices represent $X$ checks, qubits, $Z$ checks respectively.

embedding the 2D simplicial complex into $\mathbb{R}^D$ using the work of Gromov and Guth[15]. Due to the use of[4], this method only works for codes that have a sparse $\mathbb{Z}$-lift.

The concurrent work of ours on layer codes[16] provides a different approach that applies to all initial quantum codes and has a straightforward local embedding in 3D. However, it currently only works in 3D, whereas our result can be applied to any embedding dimension.

In our work, we propose a construction for almost optimal geometrically local codes. We introduce a simple square complex construction from a quantum code, avoiding the step from code to manifold. By subdividing the square complex, we are able to obtain a new code with a geometrically local embedding in $\mathbb{R}^D$. Comparing to previous works, our construction starts from arbitrary good quantum LDPC codes and can be applied to arbitrary dimensions, broadening the generality of our result, as shown in Table 1. Besides the importance of the result itself, our approach of constructing the almost optimal codes also includes several interesting observations with possible applications in other problems in coding theory. We summarize the comparison of the results in the following table.

## Results

In this paper, we introduce a construction of optimal geometrically local codes in any dimension from arbitrary good quantum LDPC codes, solving a long-term open problem. Our construction simplifies the previous work by Portnoy, and also benefits from its generality, as we do not have any additional requirement for the good quantum LDPC codes. The major advancement is built on the fact that we can construct 2D geometric complexes from codes. These complexes admit geometric operations that can be translated back to the language of codes, and we believe the construction can find other applications in coding theory.

Our main result can be stated through the following theorem:

**Theorem 1.** Given any family of quantum LDPC codes with asymptotically linear dimension, linear distance, there exists a family of $D$-dimensional geometrically local quantum codes with code

dimension $k = \widetilde{\Omega}(n^{\frac{D-2}{D}})$, code distance $d = \widetilde{\Omega}(n^{\frac{D-1}{D}})$. Moreover, if the underlying complex of code family has constant (small-set) (co) boundary expansion at level 1, the new code family has energy barrier $\mathcal{E} = \widetilde{\Omega}(n^{\frac{D-2}{D}})$, where we omit polylog $(n)$ factors in the notation $\widetilde{\Omega}$.

We will specify the expansion requirement in the Methods section, and all of the known constructions of good LPDC codes satisfy the expansion requirement[3,17].

Our method surpasses previous constructions in its generality, accommodating arbitrary quantum LDPC codes in arbitrary dimensions. Our constructions also provide alternative direct approaches to almost optimal geometrically local codes, bringing benefits to implementations. Our construction meets the distance bound $d \le O(n^{\frac{D-1}{D}})$ by Bravyi and Terhal[9], and the distance-dimension bound $kd^{\frac{2}{D-1}} \le O(n)$ in[10] upto polylog factors. We remark that the resulting code family also admits a distance-rate tradeoff indicated by the BPT bound[10] where $d = \widetilde{\Omega}((n/k)^{\frac{D-1}{2}})$ and $\mathcal{E} = \widetilde{\Omega}((n/k)^{\frac{D-2}{2}})$ for any $k \ge \widetilde{\Omega}(n^{\frac{D-2}{D}})$ through copying.

We summarize our construction as follows. We start with a quantum LDPC code with linear dimension and distance, and we introduce a simple square complex construction from the quantum code. The square complex can be viewed as a generalization of the Tanner graph of the code. While the Tanner graph provides one-dimensional geometric information of a quantum code, our square complex additionally provides 2D information of the quantum code. We will utilize the 2D geometric interpretation of the quantum code to develop a new code using geometric operations on the square complex. Quantum CSS codes can be viewed as a 3-term chain complex, and so the natural way to geometrically manipulate them should be by treating them as a two-dimensional structure. The results of this work are built on a formalization of this intuition.

To obtain a geometrically local code, we will subdivide each square in the complex into grids. It can be shown that with an appropriate subdivision parameter, the code will have a geometrically local embedding into $\mathbb{R}^D$. Here by subdivision parameter we mean the size of the grid. An example of square subdivision from the original code is shown in Fig. 1.

In the following subsections, we will introduce our construction of geometrically local codes in detail. In the subsection CSS codes and code parameters, we will introduce necessary preliminaries on CSS codes and its parameters. In the subsection Square complex from quantum code, we will show how to obtain a 2D square complex from a quantum code, which is the main ingredient of our construction. In the subsection Geometrically local codes from subdivision, we will show how to obtain a geometrically local code by dividing the square complex and embedding it into $\mathbb{R}^D$ space.

## CSS codes and code parameters

Our construction is based on CSS codes, which correspond to chain complexes naturally. Given two classical codes $C_x$, $C_z$ represented by their parity check matrices $H_x : \mathbb{F}_2^n \to \mathbb{F}_2^{m_z}$ and $H_z : \mathbb{F}_2^n \to \mathbb{F}_2^{m_x}$ that satisfy $H_x H_z^T = 0$, we are able to define a chain complex as follows:

$$\mathbb{F}_2^{m_x} \xrightarrow{\delta_0 = H_z^T} \mathbb{F}_2^n \xrightarrow{\delta_1 = H_x} \mathbb{F}_2^{m_x}. \tag{1}$$

Many properties of the code $Q$ can be described in the language of chain complexes. The $X$ and $Z$ logical operators correspond to the code $C_x$ and $C_z$, and $X$ and $Z$ stabilizers correspond to the code $C_z^\perp$ and $C_x^\perp$. The code dimension is defined by $k = \dim C_x - \dim C_z^\perp = \dim C_z - \dim C_x^\perp$. The code distance $d = \min(d_x, d_z)$ where

$$d_x = \min_{c_x \in C_x - C_z^\perp} |c_x|, \quad d_z = \min_{c_z \in C_z - C_x^\perp} |c_z|. \tag{2}$$

$$V_0 \qquad\qquad V_1 \qquad\qquad V_2$$

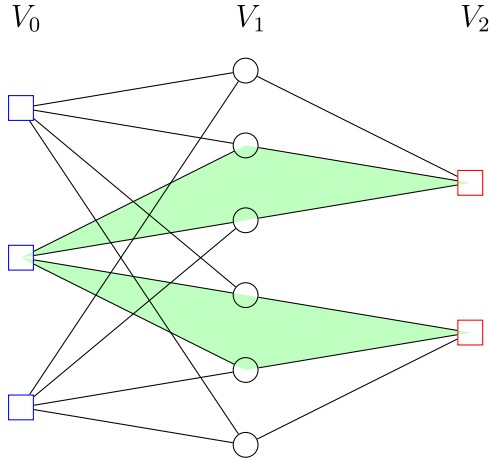

**Fig. 2 | An example of a square complex from a Tanner graph.** The blue, black, red vertices correspond to the $X$ checks, qubits, $Z$ checks respectively, we label them as $V_0$, $V_1$, $V_2$. We also include some of the face neighbors of one of the blue checks. The green squares are the faces from $S(Q)$.

We will also consider the energy barrier of our code Hamiltonian, where the energy can be evaluated by the number of violated checks in our system. The energy barrier can be defined by considering a path between two logical codewords, where each neighboring pair of strings on the path have Hamming distance 1. The energy of the path is the maximal number of violated checks of all strings on the path, and the energy barrier is the minimal energy among all paths. For example, the X energy barrier can be defined as

$$\mathcal{E}_x = \min_{\gamma_{0 \to c}, c \in C_x - C_z^\perp} \max_{c' \in \gamma_{0 \to c}} |H_x c'|, \qquad (3)$$

where $\gamma_{0 \to c}$ is the path between 0 and a codeword $c$. We can also define the Z energy barrier similarly, and define the energy barrier of the system as the minimum of the two. In the following parts of the paper, we will often adapt the language of chain complexes, and use level 0,1,2 vertices for $X$ checks, qubits, $Z$ checks, respectively.

### Square complex from quantum code

It is known that the Tanner graph of a good quantum LDPC code $Q$ cannot have a geometrically local embedding, since the code's distance violates the bound in ref. 9. The Tanner graph $\mathcal{T}(Q)$ of a quantum code is the tripartite graph with $X$ checks, $Z$ checks, and qubits as vertices, connected if a check acts on a qubit. To overcome the barrier of embedding expanders, we will show a process that geometrizes the quantum code $Q$.

The geometrization process says that for every chain complex (or quantum CSS code), there is an associated simplicial/square complex, such that for many geometric operations performed on the simplicial/square complex, such as subdivision and sliding of simplices, there are corresponding operations on the chain complex. In our construction, we will first construct a square complex from a good quantum LDPC code. Then, we subdivide the code in the same way we subdivide a square complex.

For a good quantum LDPC code $Q$, we will derive a square complex $\mathcal{S}(Q)$ from the Tanner graph $\mathcal{T}(Q)$. Since the $X$ and $Z$ stabilizers of the code $Q$ commute, in the Tanner graph $\mathcal{T}(Q)$, each pair of $X$ and $Z$ stabilizers shares an even number of common neighbor qubits. By this observation, we can pair up the common neighbor qubits, and obtain a square face set $F$ where each face $f \in F$ consists of one $X$ check, one $Z$ check, and a pair of their common neighbor qubits in the Tanner graph. We can obtain a 2D square complex $\mathcal{S}(Q) = (V, E, F)$ from the

Tanner graph $\mathcal{T}(Q) = (V, E)$ by including the additional face set $F$. Readers can refer to Fig. 2 as an example.

For good quantum LDPC codes under our consideration, we can stop here and take the square complex $\mathcal{S}(Q)$ as our 2D structure construction. For more pathological codes, we need some additional structure. The corresponding 2D structure of a general code will be a relaxed notion of a square complex, which we call the square subspace complex. A square subspace complex $\widetilde{S} = (\widetilde{V}, \widetilde{E}, \widetilde{F})$ also consists of vertices $\widetilde{V}$, edges $\widetilde{E}$, and faces $\widetilde{F}$, but it is no longer required to be downward closed, i.e. a face $f \in \widetilde{F}$ may contain an edge that is not in $\widetilde{E}$. Additionally, we require the face structure to have a connected link around the level 0 and 2 vertices. The link of a vertex $v$ is a graph with 'vertices' as its neighboring edges $e$ and 'edges' induced by its neighboring faces $f$ which connects the two 'vertices' $v \in e \in f, v \in e' \in f$. This condition automatically holds for good quantum LDPC codes. The detailed construction of the square subspace complex for general codes can be found in the supplementary material. We will summarize our quantum code to 2D complex procedure in the following theorem, which might be of independent interest.

**Theorem 2.** Given an arbitrary bounded degree 3-term chain complex $X$ over $\mathbb{F}_2$, there exists an embedding to a bounded degree square subspace complex $I_{\text{code} \to \text{square}} : X \to \widetilde{S}(X)$. The square subspace complex $\widetilde{S}(X)$ has the same vertices and edges as the Tanner graph of $X$. It has a face structure where the link of its level 0 and level 2 vertices is connected.

We will not emphasize the difference between $\mathcal{S}(Q)$ and $\widetilde{\mathcal{S}}(Q)$ in the exposition of our main result. For simplicity, we will use $\mathcal{S}(Q)$ for our square complex.

### Geometrically local codes from subdivision

Now we subdivide the structure $\mathcal{S}(Q)$ to obtain a Tanner graph $\mathcal{T}(Q_L)$. Taking inspiration from the surface code construction[4], it is natural to subdivide each square face in $\mathcal{S}(Q)$ to an $L \times L$ size grid and assign $X$ checks, $Z$ checks, qubits correspondingly as in surface codes. For each subdivided face, we define the level 0, level 1, and level 2 vertices of the new chain complex $X_L$ as follows, under the convention that coordinates increase as going right and up in Fig. 1:

- $X_L(0)$ are the set of vertices with $i, j$ both being even.
- $X_L(1)$ are the set of vertices with $i, j$ one being even, one being odd.
- $X_L(2)$ are the set of vertices with $i, j$ both being odd.

The corresponding vertices are colored blue, black, and red in Fig. 1.

For the code $Q_L$, we have the following theorem for its parameters:

**Theorem 3.** Given a good qLDPC code $Q$ with parameter $[[n, k = \Theta(n), d = \Theta(n)]]$, the new code $Q_L$ has parameter $[[n_L = \Theta(nL^2), k_L = k, d_L = \Theta(dL)]]$. Moreover, if $X$ has constant (small-set) (co)boundary expansion at level 1, the new code $Q_L$ has energy barrier $\mathcal{E}_L = \Theta(\mathcal{E}) = \Theta(n)$.

After subdivision with large enough $L$, the Tanner graph $\mathcal{T}(Q_L)$ has a geometrically local embedding in $\mathbb{Z}^D$ provided by the following theorem from[13].

**Theorem 4.** For any $L$-subdivided square complex $\mathcal{S}_L = (V_L, E_L, F_L)$ from a square complex $\mathcal{S} = (V, E, F)$, there exists an embedding map $I_{\text{square}_L \to \text{euclid}} : V_L \to \mathbb{Z}^D$ with constant $a, b = \Theta(1)$ such that for $L = \Theta(|V|^{\frac{1}{D-2}} \text{polylog} |V|)$:

1. Geometrically local: For all adjacent vertices on the complex $\{v_0, v_1\} \in E_L$, the distance between corresponding points in $\mathbb{Z}^D$ is bounded, i.e. $|I_{\text{square}_L \to \text{euclid}}(v_0) - I_{\text{square}_L \to \text{euclid}}(v_1)| \leq a$.
2. Bounded density: The number of vertices at each point in $\mathbb{Z}^D$ is bounded, i.e. $\forall x \in \mathbb{Z}^D, |I_{\text{square}_L \to \text{euclid}}^{-1}(x)| \leq b$.

Setting $L = \Theta(|V|^{\frac{1}{D-2}} \text{polylog} |V|) = \Theta(n^{\frac{1}{D-2}})$, and applying the construction to the code $Q_L$, we can obtain Theorem 1.

## Discussion

We expect our construction to find more applications in both algebraic topology and coding theory. We point out several possible directions for future work.

The first direction is to geometrize and embed longer chain complexes. This work focuses on quantum CSS codes, which are 3-term chain complexes over $\mathbb{F}_2$, and where we endow them with 2D geometrical structures. We expect a similar result holds in higher dimensions over arbitrary finite fields or $\mathbb{Z}$. In particular, we can endow a $t+1$-term chain complex with a $t$-dimensional CW complex structure after pairing. Furthermore, when the chain complex has bounded degree with bounded entry values, the CW complex also has bounded degree. When combined with the embedding theorem for simplicial complexes, this provides a way to embed a $t+1$-term chain complex in $\mathbb{R}^D$ with $D > t$, which likely saturates the generalization of the BPT bound.

Our construction can also shed light on weight reduction of quantum codes. Given a qLDPC code, can one convert it into another qLDPC code with similar properties while having a smaller check weight? This problem was first answered by Hastings in ref. [18]. The intuition behind the work is elegantly articulated in the introduction of the paper. The reasoning is that there seem to be a general procedure that transforms quantum codes into manifolds. (Note that the reverse process is straightforward, as manifolds can always be converted into quantum codes.) Given that manifolds can naturally be refined such that each cell is only attached to a constant number of other cells, this suggests a method of weight reduction.

However, a challenge in implementing this idea is that the procedure for transforming a code into a manifold, as described in ref. [14], does not work for all codes. Nevertheless, Hastings found an alternative approach to achieve weight reduction, as detailed in ref. [18].

We believe that given our procedure to endow an arbitrary quantum code with a 2D geometrical structure, we can fully realize Hastings' intuition. We will describe the details in future work.

**Remark 5**. After the acceptance of the paper, we are aware of a new result[19] that can remove the additional polylog factor in Theorem 4. The result can improve our result to an asymptotically optimal family of geometrically local quantum codes.

## Methods

In this section, we will introduce the methods we used in proving Theorem 3. Instead of giving every detail of the proof, we will give the outline of our proof, summarize the key ideas, and leave the full proof in the supplementary material for interested readers.

Note that in our construction, the $X$ check and $Z$ check vertices are symmetric, that is if we swap the view of $X$ check and $Z$ check vertices, the proof of code properties in the two cases are essentially the same. Therefore, in our proof, we will only focus on proving one direction, i.e. the coboundary direction.

To analyze the properties of the new code $Q_L$, we relate it with the original code $Q$. Note that every CSS code has a corresponding chain complex. We use $X$ to denote the chain complex of the qLDPC code $Q$ and $X_L$ for our target subdivision code $Q_L$.

### Code size

The following estimation on the size of our new chain complex $X_L$ turns out to be useful in estimating the size of our new code:

**Claim 6**. Given the maximum degree Emphasis>/Emphasis> of the chain complex $X$, we have the following bound for the complex $X_L$

$$\frac{L^2}{4}|X(i)| \le |X_L(i)| \le \frac{\Delta^2 L^2}{2}(|X(0)| + 2|X(i)| + |X(2)|). \quad (4)$$

In particular, if $\Delta = \Theta(1)$, we have $|X_L(i)| = \Theta(L^2|X(i)|)$.

The claim can be verified by counting the number of faces in our chain complex construction.

### Code dimension

For the ease of our analysis of other code parameters, we introduce a chain map $\mathcal{F}$ from $X : \mathbb{F}_2^{X(0)} \to \mathbb{F}_2^{X(1)} \to \mathbb{F}_2^{X(2)}$ to $X_L : \mathbb{F}_2^{X_L(0)} \to \mathbb{F}_2^{X_L(1)} \to \mathbb{F}_2^{X_L(2)}$ shown in the following commutative diagram. Furthermore, we will show that $\mathcal{F}$ induces an isomorphism on the relevant (co)homology groups used for our dimension arguments.

$$
\begin{array}{ccccc}
\mathbb{F}_2^{X(0)} & \xrightarrow{\delta_0} & \mathbb{F}_2^{X(1)} & \xrightarrow{\delta_1} & \mathbb{F}_2^{X(2)} \\
\downarrow{\mathcal{F}_0} & & \downarrow{\mathcal{F}_1} & & \downarrow{\mathcal{F}_2} \\
\mathbb{F}_2^{X_L(0)} & \xrightarrow{\delta_0} & \mathbb{F}_2^{X_L(1)} & \xrightarrow{\delta_1} & \mathbb{F}_2^{X_L(2)}
\end{array} \quad (5)
$$

This chain map $\mathcal{F}$ will relate the codewords of $Q(X)$ to the codewords of $Q(X_L)$. It will also play an important role when we are studying the properties of the code $Q(X_L)$. For the convenience of describing the chain map, we will first define some regions in the faces $\widetilde{F}$:

- $S$ contains vertices with $0 \le i, j \le L - 1$,
- $T$ contains vertices with $0 \le i \le L - 1, j = L$ or $0 \le j \le L - 1, i = L$.
- $U$ contains the vertex $i = j = L$.

Now we define the maps $\mathcal{F}_i : \mathbb{F}_2^{X(i)} \to \mathbb{F}_2^{X_L(i)}, i \in \{0, 1, 2\}$, which will be used repeatedly in our analysis:

- Given $\widetilde{c}_0 \in \mathbb{F}_2^{X(0)}$, we define $\mathcal{F}_0(\widetilde{c}_0)$ by repeating the value $\widetilde{c}_0(v_0)$ at each component $S_{v_0}$ corresponding to $v_0 \in X(0)$, including the level 0 dummy faces. We set the level 2 dummy faces to 0.
- Given $\widetilde{c}_1 \in \mathbb{F}_2^{X(1)}$, we define $\mathcal{F}_1(\widetilde{c}_1)$ by repeating the value $\widetilde{c}_1(v_1)$ at each component $T_{v_1}$ corresponding to $v_1 \in X(1)$, and the values in other part to 0.
- Given $\widetilde{c}_2 \in \mathbb{F}_2^{X(2)}$, we define $\mathcal{F}_2(\widetilde{c}_2)$ by setting the value $\widetilde{c}_2(v_2)$ at the corresponding vertex $U_{v_2}$, and 0 elsewhere.

The readers can refer to Fig. 3 for a figurative explanation of the regions.

In order to simplify our further discussion, Fig. 4 provides two examples of connected components of the regions $S$ and $T$, respectively.

We will verify that the $\mathcal{F}$ we constructed above is a homotopy equivalence in the Supplementary materials.

Using this fact, we can show that the new code has the same dimension as the old code, $k$.

### Distance and energy barrier

We will analyze the distance and energy barrier of the code in the language of the chain complex. To finish proving Theorem 3, we will need to introduce the following expansion notions of a chain complex.

**Definition 7**. (Small-Set (Co)Boundary Expansion) We say that $X : \mathbb{F}_2^{X(0)} \xrightarrow{\delta_0} \mathbb{F}_2^{X(1)} \xrightarrow{\delta_1} \mathbb{F}_2^{X(2)}$ is an $(\alpha, \beta)$-small-set boundary expander if

$$
\begin{aligned}
\forall c_1 &\in \mathbb{F}_2^{X(1)}, |c_1| \le \alpha|X(1)| : \\
&\exists c_2 \in \mathbb{F}_2^{X(2)}, \beta|c_1 + \partial_2 c_2| \le |\partial_1 c_1|.
\end{aligned} \quad (6)
$$

Similarly, $X$ is an $(\alpha, \beta)$-small-set coboundary expander if

$$
\begin{aligned}
\forall c_1 &\in \mathbb{F}_2^{X(1)}, |c_1| \le \alpha|X(1)| : \\
&\exists c_0 \in \mathbb{F}_2^{X(0)}, \beta|c_1 + \delta_0 c_0| \le |\delta_1 c_1|.
\end{aligned} \quad (7)
$$

By the symmetry between the $X$ and $Z$ check vertices in our construction, in the following parts, we will only prove the coboundary

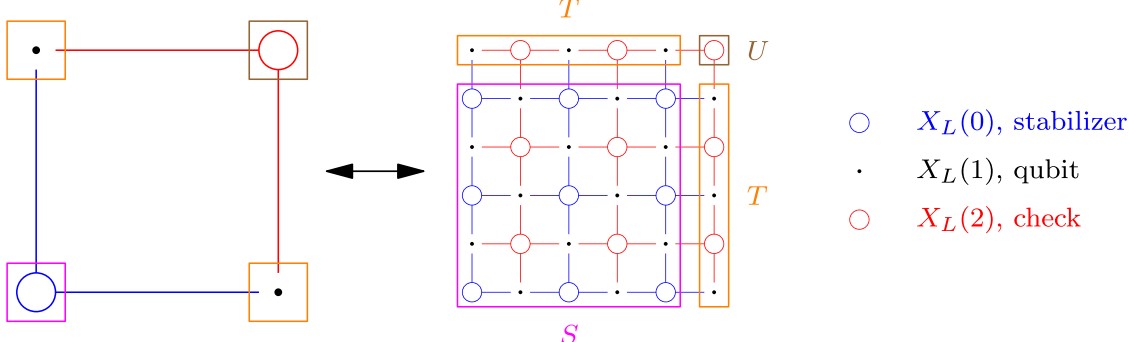

**Fig. 3 | The figure describes regions *S*, *T*, *U* used in the chain map.** Note that the connected regions have a natural bijection with the vertices in chain complex *X*, as we use the same color to identify the bijection.

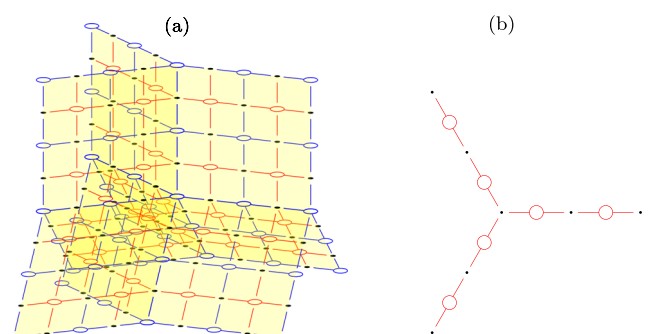

**Fig. 4 | Example of the local structure of the regions *S* and *T*. a** is an example of region *S*, and (**b**) is an example of region *T*.

expansion properties of the code, as the boundary expansion proof will be the same.

We can relate the distance and energy barrier of a code $Q(X_L)$ to the small set (co)boundary expansion parameters of its corresponding complex $X_L$ through the following theorem:

**Theorem 8.** If the complex $X_L$ has $(\alpha, \beta)$-small-set boundary and coboundary expansion, the corresponding code has distance $d > \alpha|X_L(1)|$ and energy barrier $\mathcal{E} \geq \alpha\beta|X_L(1)|$.

**Proof..** Proof of the theorem is a translation between languages. For any nontrivial $X$ codewords $c_1 \in \mathbb{F}_2^{X_L(1)}$, we have that $\delta_1 c_1 = 0$, thus if $|c_1| \leq \alpha|X_L(1)|$, by the definition of small-set coboundary expansion, there exists some $c_0 \in \mathbb{F}_2^{X_L(0)}$ such that $|c_1 + \delta_0 c_0| = 0$, meaning that $c_1$ is some stabilizer, a contradiction.

For the energy barrier, consider some minimal weight $c_1$ in its equivalence class $c_1 + B^1(X_L)$ with weight $|c_1| = \alpha|X_L(1)|$, thus by the definition of small-set coboundary expansion, we have that $\beta|c_1| \leq \beta|c_1 + \delta_0 c_0| \leq |\delta_1 c_1|$, implying that the energy barrier is at least $\alpha\beta|X_L(1)|$. 7D2

To obtain the small-set coboundary expansion parameters of the chain $X_L$, we will first analyze the local coboundary expansion properties of the areas $S$ and $T$, and we connect the global coboundary expansion of $X_L$ back to the expansion property of $X$ with the chain map $\mathcal{F}$. To define the local coboundary expansion properties of the chain complex, we will use the definition of a chain complex with a boundary. The chain complex $Y : \mathbb{F}_2^{Y(0)} \xrightarrow{\delta_0} \mathbb{F}_2^{Y(1)} \xrightarrow{\delta_1} \mathbb{F}_2^{Y(2)}$ is extended to have the additional boundary structure $Y_\partial$, $Y \cup Y_\partial : \mathbb{F}_2^{Y(0) \cup Y_\partial(0)} \xrightarrow{\delta'_0} \mathbb{F}_2^{Y(1) \cup Y_\partial(1)} \xrightarrow{\delta'_1} \mathbb{F}_2^{Y(2) \cup Y_\partial(2)}$,

Let us first define the following coboundary expansion properties of the local chain complexes with boundaries:

**Definition 9.** A chain complex $Y$ with boundary $Y_\partial$ is a $(\beta_i, \eta_i)$-coboundary expander at level $i$ if for all $\widehat{f}_i \in \mathbb{F}_2^{Y(i)}$, there exists $f_i \in \widehat{f}_i + B^i \subset \mathbb{F}_2^{Y(i)}$ such that

$$(1) \ |f_i|_{\text{int}} \leq |\widehat{f}_i|_{\text{int}}, \quad (2) \ \beta_i|f_i|_{\text{int}} \leq |\delta_i\widehat{f}_i|_{\text{int}}, \quad (3) \ \eta_i|\delta_i f_i|_\partial \leq |\delta_i f_i|_{\text{int}}. \tag{8}$$

where $|\cdot|_{\text{int}}$ is the Hamming weight of the vector restricted to the space $\mathbb{F}_2^{Y(i)}$, and $|\cdot|_\partial$ is the Hamming weight of the vector restricted to the space $\mathbb{F}_2^{Y_\partial(i)}$.

We can relate local expansion to global expansion through the following theorem:

**Theorem 10.** If $X$ has $(\alpha_{\text{qLDPC}}, \beta_{\text{qLDPC}})$-small-set coboundary expansion with $\alpha_{\text{qLDPC}}, \beta_{\text{qLDPC}} = \Theta(1)$, and for each local complex $S$ and $T$, they are $(\beta_i, \eta_i)$-coboundary expanders with $\beta_i = \Theta(1/L)$, $\eta_i = \Theta(1)$ (for $S$ we take $i = 0, 1$, and for $T$ we only consider $i = 0$), we have that the complex $X_L$ has $(\alpha, \beta)$-small-set coboundary expansion with $\alpha, \beta = \Theta(1/L)$. Moreover, $\alpha$ is only determined by $\alpha_{\text{qLDPC}}$.

The proof idea of the theorem is to use some local error correction process on $S$ and $T$ to reduce the code of the complex $X_L$ back to some original code in the complex $X$ via the chain map. The local error correction process is determined by the coboundary expansion of the regions $S$ and $T$, which bears some similarity to surface codes and repetition codes. We will bound the expansion of $X_L$ using the expansion of $X$.

By Theorem 8, we can obtain the desired code parameters in Theorem 3. We will then turn to prove the local expansion parameters of $S$ and $T$ as stated in Theorem 10.

**Theorem 11.** Every local complex $S$ with the boundary $S_\partial$ is a $(\beta_0^S, \eta_0^S)$-coboundary expander at level 0, and every local complex $T$ with boundary $T_\partial$ is a $(\beta_0^T, \eta_0^T)$-coboundary expander at level 0. We have that $\beta_0^S, \beta_0^T = \Theta(1/L)$, $\eta_0^S, \eta_0^T = \Theta(1)$.

**Theorem 12.** Every local complex $S$ with the boundary $S_\partial$ is a $(\beta_1^S, \eta_1^S)$-coboundary expander at level 1. We have that $\beta_1^S = \Theta(1/L)$, $\eta_1^S = \Theta(1)$.

The proof of these theorems can be found in the supplementary materials.

## Data availability

There was no data generated during this work.

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

## Acknowledgements

X.L. is supported by National Key Research and Development Program of China (Grant No. 2023YFA1009403), National Natural Science Foundation of China (Grant No. 12347104), and Beijing Science and Technology Planning Project (Grant No. Z25110100810000). T.C.L. is supported in part by funds provided by the U.S. Department of Energy (D.O.E.) under cooperative research agreement DE-SC0009919, and by the Simons Collaboration on Ultra-Quantum Matter, which is a grant from the Simons Foundation (652264, JM). A.W. supported by the MIT Department of Physics and by the U.S. Department of Energy, Office of Science, National Quantum Information Science Research Centers, Quantum Systems Accelerator.

## Author contributions

T.C.L., A.W. and M.H.H. constructed the first optimal geometrically local QLDPC codes in any dimension under a restricted local product construction. X.L., T.C.L. and M.H.H. removed the restriction.

## Competing interests

The authors declare no competing interests.
