## [Transparent Peer Review file · Nature Communications]

Almost Optimal Geometrically Local Quantum LDPC Codes in any Dimension

Corresponding Author: Dr Min-Hsiu Hsieh

Version 0:

Reviewer comments:

Reviewer #1

(Remarks to the Author)

The paper proposes a generic pipeline that turns any good CSS qLDPC code (linear rate and distance) into a D -dimensional geometrically-local code by (i) extracting a 2-D “square (subspace) complex” from the Tanner graph via pairing common X/Z neighbors, (ii) subdividing each square into an $L \times L$ grid, and (iii) invoking an embedding theorem to place the subdivided complex in Z^D with bounded interaction range.

If fully correct, the result is very important. It generalizes previous near-optimal locality constructions (Portnoy; Williamson–Baspin’s 3-D layer codes) by removing specialized prerequisites and working in any D , while meeting the classic BPT constraints asymptotically (modulo the upcoming result by Lin & Portnoy).

I believe that the main result will matter across quantum coding and condensed-matter realizations, and the “code \rightarrow 2D complex \rightarrow operations \rightarrow back to code” paradigm looks very promising and reusable. The core idea is elegant, and the construction is conceptually clear. The manuscript is generally well organized, with a logical flow from the problem statement to the results and proofs.

Unfortunately, the manuscript contains several unproven statements and one formally incorrect statement about the authors’ prior work [14], which, in my opinion, prevents publication in its current form.

1. The abstract claims “we introduce a construction that can transform *any* good quantum LDPC code into an optimal geometrically local quantum code,” but the main text (Theorem II.1) weakens this to families with linear dimension, linear distance, and *a linear energy barrier*, adding an assumption not stated in the abstract. More importantly, as I can see, the proofs further require *small-set (co)boundary expansion* of the seed good qLDPC code; the paper does not derive this from having large distance and/or a macroscopic energy barrier, and I am not aware of any result that would justify such an implication. I think that it is essential to add a complete, explicit proof of Theorem II.3 (e.g., in the supplementary material) and to revise its statement so that it matches the abstract exactly.

2. The manuscript also states (lines 196–199) that the Tanner graph of every good qLDPC code is an expander; I have not seen this theorem before, and a precise citation (or a corrected statement) is needed here.

3. The authors write: “In the earlier version of the paper [14], we showed that the step from code to manifold in Portnoy’s work can be avoided if we start from *balanced product codes*.” First, Portnoy’s work arXiv:2012.09271 does not use balanced product codes. It is stated explicitly in Portnoy’s paper that the construction there is based on lifted product codes and their modification – quantum Tanner codes. In [14, p. 10], the authors present a construction $T(\text{Cay}(A, G), CA) \otimes_G T(\text{Cay}(G, B), CB)$, based on the tensor product of Tanner codes on Cayley graphs (apparently implicitly assuming subsystem balanced product codes, which were added in the second version of arXiv:2012.09271), and claim – citing arXiv:2111.03654 – that these are lifted product codes.

However, this identification is incorrect, and the formula given there is false. Moreover, for the codes $T(\text{Cay}(A, G), CA) \otimes_G T(\text{Cay}(G, B), CB)$, the local views do not have a product structure – a property required for the construction in [14] to work. For example, if you look at a 2-cell $\{e, es, s^{-1}e, s^{-1}es\} = \{e, s, s^{-1}\}$, where s is in $A = B = S = S^{-1}$, this is not a square (only 3 vertices!), which means that $\text{Cay}(A, G) \times_G \text{Cay}(G, B)$ is not even a square complex according to the definition

in the present manuscript.

It appears that the authors in [14] have confused two *different and inequivalent* frameworks that both use tensor products: Lifted product codes (arXiv:1904.02703, arXiv:2012.04068), i.e. *CSS codes* obtained by combining codes defined using G -lifts of bipartite graphs such as *Zemor codes*, whose Tanner graphs are lifts of Cartesian products of graphs and thus by definition have a local product structure (all such complexes can be obtained using twisted product from graph theory). Balanced product codes (arXiv:2012.09271), which are *subsystem codes* obtained by combining *Tanner codes on Cayley graphs*, such as $\text{Cay}(S, G) \times_G \text{Cay}(G, S)$ from [14, p. 10]. These are not lifts of graphs and they do *not* have a local product structure.

I stress that this local product structure was not only used in [14] but is in fact a cornerstone of all known constructions of good qLDPC codes. Balanced product codes do not possess this property. Therefore, it is highly misleading and formally incorrect to claim that balanced product codes have this structure and to state that they were used in [14] to obtain geometrically local codes.

My recommendation: major revision. The core idea is strong and, with the clarifications/fixes above, this can be a very impactful paper. I view the needed changes as addressable:

1. tighten the formal statements to match what is actually proved (or add the missing proofs),
2. correct and clarify the statements in lines 196–199 about expander graphs and in lines 78, 553 about balanced product codes,
3. clarify a handful of language/typo issues mentioned below.

Additional comments to the authors

1. You use simultaneously “vertice” and “vertex” in many places, which is not very good. Please, try to choose one version and stick to it.
2. You also use “energy barrier” (in the main text) and “energy gap” (in the supplementary)
3. I would suggest replacing repeated “we would ...” with “we ...” or “we will ...” (present or simple future). Examples: “we would use the following embedding result” → “we use the following embedding result”; “we would first verify ...” → “we first verify ...”, “would also consider” → “we also consider”.
4. [31]: qubits and check → qubits and checks
5. In FIG. 2: one of the blue check → one of the blue checks
6. [118]: alterative → alternative
7. [290]: stray punctuation (“...from [13, 16], .”), please fix.
8. [401]: “qLPDC” → qLDPC
9. [416, 839]: commutation diagram → commutative diagram
10. FIG. 4: Remove space before period: “...region T .” → “...region T.”
11. [464]: Similarly, X is a (α, β, γ) -small-set → Similarly, X is an (α, β, γ) -small-set
12. [487]: in its equivalent class → in its equivalence class
13. In FIG. 3 caption: “FIG. 3: Regions S, T, U , note that the connected regions have a natural bijection...” → “FIG. 3: Regions S, T, U . Note that the connected regions ...”.
14. [688]: chain → cochain (see, also [702]); if you want to simultaneously use both direction it is better to say (co)chain
15. [709]: $\text{eps}_x(c_x) = \dots$ → $\text{eps}_x(c) = \dots$
16. [740]: L subdivided complex → L -subdivided complex; also in FIG. A.5: L -subdivision, and “face f .(Figure 5 in [14])” → “face f . (Figure 5 in [14]).” (space after the period!)
17. [766]: a X -stabilizer → an X -stabilizer
18. [780]: “every check v_2 that share a qubit” → “every check v_2 that shares a qubit.”
19. [782]: the endpoints of the ‘edges’ corresponds → the endpoints of the ‘edges’ correspond
20. [804]: downward close → downward closed
21. [811]: you refer to the “brown” face, but there are no brown faces in Fig. 2
22. [823]: there is at most → there are at most
23. The methods section states F is “a homotopy equivalence between two chain complexes.” The supplement (as excerpted) proves commutativity and that F_{-1} induces a bijection on Z^1/B^1 (i.e., preserves the logical space). That is sufficient for Theorem II.3’s claim $k_L = k$, but it falls short of a full chain homotopy equivalence unless additional chain homotopies are constructed at other degrees. Please either (a) provide the missing homotopies, or (b) soften the claim to “ F induces isomorphism on the relevant (co)homology groups used for k .”
24. [848]: between the region → between the regions
25. [888]: take the same value → takes the same value
26. [923]: there exist → there exists
27. [976]: One easily check that → One easily checks that
28. [1071] left hand → left-hand

Reviewer #2

(Remarks to the Author)

This is a nicely presented work that provides an incremental extension of previous work by some of the same authors, following an earlier work by E. Portnoy. The main result is a mapping from arbitrary good qLDPC codes to local codes on finite dimensional lattices with (almost) the best possible code parameters that are achievable for such codes. I have included the qualifier almost because the main result of saturating optimal parameter scaling for local codes in any dimension, rather than achieving this only up to $1/\text{polylog}$ factors, appears to depend on an unpublished work cited as Ref.16. There was no clear explanation of how this improved result is obtained beyond this citation. The main innovation of this work seems to lie in generalizing the family of codes that the construction applies to. As the authors point out, this offers promise of potential future applications including weight reduction of quantum codes. I tend to agree with the authors that this work is likely to lead to interesting follow ups, but I do not think it crosses the line for acceptance to Nature Communications in its current form given that it does not contain proof that it does indeed improve on the code properties obtained in previous works.

Reviewer #3

(Remarks to the Author)

The authors present a generalization of their earlier work [14], proposing a method to construct optimal geometrically local quantum codes from arbitrary good quantum LDPC codes. Compared to prior constructions, this approach offers improved generality as it applies to arbitrary spatial dimensions and is not limited to specific code families.

A particularly interesting aspect of the work is the possibility that one could convert a quantum LDPC code into another qLDPC code with similar properties but potentially reduced check weight, which could provide practical advantages. However, the extent of the improvement remains unclear, as the manuscript does not currently include concrete examples or performance evaluations to support or illustrate the practical advantages of the construction.

There are two points that could be strengthened to evaluate the significance to the field:

1) The paper does not provide explicit examples of geometrically local quantum codes generated using the proposed method. Including such examples with numerical simulations, demonstrating decoding performance, would allow for comparisons with existing solutions such as bivariate bicycle codes or surface codes. Metrics such as overhead in terms of physical qubits and logical error rate under standard noise models would be particularly useful.

2) While the paper emphasizes geometric locality, it would be valuable to discuss how the construction might apply in non-local architectures, such as those incorporating c-couplers or photonic interconnects that support long-range qubit interactions. Clarifying whether the method still offers advantages in such settings, or how it could be adapted, would broaden its relevance.

Version 1:

Reviewer comments:

Reviewer #1

(Remarks to the Author)

I have read the revised version of the manuscript. The authors have addressed my previous comments, and the presentation is now much better. I have no further remarks or concerns at this stage.

Reviewer #2

(Remarks to the Author)

Given the clarification and changes, I now find this result suitable for publication provided the following typos are fixed:

Optimal -> almost optimal in the following:

-“In the current version of our work, we generalized the idea in [14] to construct optimal geometrically local codes.”

-Line 90 “constructing the optimal codes”

-Line 127 ...

-Line 667 ...

We believe our manuscript has benefited substantially from the insightful comments provided in the previous review round. We sincerely thank all of the reviewers for their suggestions and comments.

We would like to clarify that this submission is a merge of [1, 2], and the first work is not going to be published elsewhere.

In the attached response letter, we provided a point-to-point response to the reviewers' concerns, along with detailed modifications of our manuscript. The major revision includes:

1. *Changes in the title and main theorem on optimality.* Multiple reviewers have raised concerns about our reference to an unpublished work [3]. After private communications with the authors, we learned that the manuscript is still under preparation and cannot be published online in the near future. We have changed the embedding theorem using the result of [3] to the theorem shown in [4] with extra polylog loss. Due to the adjustment in the embedding theorem, our main theorem will introduce a polylog loss in the asymptotic rate and distance, and we have changed our claim from optimal to almost optimal accordingly. Please refer to the response to Reviewer 2 for more details.
2. *Structural adjustment of our main theorem.* We realize our previous statement of the main theorem will cause confusion that we need linear rate, distance, and energy barrier to obtain an almost optimal geometrically local code. In our current statement, we have separated the theorem on optimal geometrically local codes and its energy barrier, as we only require linear rate and distance to obtain an almost optimal geometrically local code. And on the optimal energy barrier, we require a boundary expansion of the underlying complex. Please refer to the response to Reviewer 1 for more details.

Other noticeable revisions include:

1. Removal of several unproven claims. These claims only serve for the purpose of providing intuition, and do not hurt any of our theorems.
2. More accurate naming of the primitives to avoid confusion. For example, we changed “balance product codes” to “codes with a local product structure”.
3. Multiple typos fixed.

Reviewer #1

The paper proposes a generic pipeline that turns any good CSS qLDPC code (linear rate and distance) into a D-dimensional geometrically-local code by (i) extracting a 2-D “square (subspace) complex” from the Tanner graph via pairing common X/Z neighbors, (ii) subdividing each square into an $L \times L$ grid, and (iii) invoking an embedding theorem to place the subdivided complex in \mathbb{Z}^D with bounded interaction range.

If fully correct, the result is very important. It generalizes previous near-optimal locality constructions (Portnoy; Williamson–Baspin’s 3-D layer codes) by removing specialized prerequisites and working in any D, while meeting the classic BPT constraints asymptotically (modulo the upcoming result by Lin & Portnoy).

I believe that the main result will matter across quantum coding and condensed-matter realizations, and the “code \rightarrow 2D complex \rightarrow operations \rightarrow back to code” paradigm looks very promising and reusable. The core idea is elegant, and the construction is conceptually clear. The manuscript is generally well organized, with a logical flow from the problem statement to the results and proofs.

Unfortunately, the manuscript contains several unproven statements and one formally incorrect statement about the authors’ prior work [14], which, in my opinion, prevents publication in its current form.

1. The abstract claims “we introduce a construction that can transform *any* good quantum LDPC code into an optimal geometrically local quantum code,” but the main text (Theorem II.1) weakens this to families with linear dimension, linear distance, and *a* linear energy barrier, adding an assumption not stated in the abstract. More importantly, as I can see, the proofs further require *small-set (co)boundary expansion* of the seed good qLDPC code; the paper does not derive this from having large distance and/or a macroscopic energy barrier, and I am not aware of any result that would justify such an implication. I think that it is essential to add a complete, explicit proof of Theorem II.3 (e.g., in the supplementary material) and to revise its statement so that it matches the abstract exactly.

2. The manuscript also states (lines 196–199) that the Tanner graph of every good qLDPC code is an expander; I have not seen this theorem before, and a precise citation (or a corrected statement) is needed here.

3. The authors write: “In the earlier version of the paper [14], we showed that the step from code to manifold in Portnoy’s work can be avoided if we start from *balanced product codes*.” First, Portnoy’s work arXiv:2012.09271 does not use balanced product codes. It is stated explicitly in Portnoy’s paper that the construction there is based on lifted product codes and their modification – quantum Tanner codes. In [14, p. 10], the authors present a construction $T(\text{Cay}(A, G), C_A) \otimes_G T(\text{Cay}(G, B), C_B)$, based on the tensor product of Tanner codes on Cayley graphs (apparently implicitly assuming subsystem balanced product codes, which were added in the second version of arXiv:2012.09271), and claim – citing arXiv:2111.03654 – that these are lifted product codes.

However, this identification is incorrect, and the formula given there is false. Moreover, for the codes $T(\text{Cay}(A, G), C_A) \otimes_G T(\text{Cay}(G, B), C_B)$, the local views do not have a product structure – a property required for the construction in [14] to work. For example, if you look at a 2-cell $\{e, es, s^{-1}e, s^{-1}es\} = \{e, s, s^{-1}\}$, where s is in $A = B = S = S^{-1}$, this is not a square (only 3 vertices!), which means that $\text{Cay}(A, G) \times_G \text{Cay}(G, B)$ is not even a square complex according to the definition in the present manuscript.

It appears that the authors in [14] have confused two *different and inequivalent* frameworks that both use tensor products: Lifted product codes (arXiv:1904.02703, arXiv:2012.04068), i.e. *CSS codes* obtained by combining codes defined using G-lifts of bipartite graphs such as *Zemor codes*, whose Tanner graphs are lifts of Cartesian products of graphs and thus by definition have a local product structure (all such complexes can be obtained using twisted product from graph theory). Balanced product codes (arXiv:2012.09271), which are *subsystem codes* obtained by combining *Tanner codes on Cayley graphs*, such as $\text{Cay}(A, G) \times_G \text{Cay}(G, B)$ from [14, p. 10]. These are not lifts of graphs and they do *not* have a local product structure.

I stress that this local product structure was not only used in [14] but is in fact a cornerstone of all known constructions of good qLDPC codes. Balanced product codes do not possess this property. Therefore, it is highly misleading and formally incorrect to claim that balanced product codes have this structure and

to state that they were used in [14] to obtain geometrically local codes.

My recommendation: major revision. The core idea is strong and, with the clarifications/fixes above, this can be a very impactful paper. I view the needed changes as addressable:

1. tighten the formal statements to match what is actually proved (or add the missing proofs),
2. correct and clarify the statements in lines 196–199 about expander graphs and in lines 78, 553 about balanced product codes,
3. clarify a handful of language/typo issues mentioned below.

Author’s Reply: We thank the reviewer for his detailed review and for pointing out multiple problems in our manuscript. We also thank the reviewer for acknowledging the merits of our paper.

For the reviewer’s first concern, we would like to clarify that to achieve an almost optimal geometrically local code, we only require the base code to have linear rate and distance. We additionally require small-set (co)boundary expansion when we prove the bound on the energy barrier. In the revised statement of Theorem II.1 and II.3, we have decoupled the results on rate and distance from the result on energy barrier, and we also adjusted the statement of Theorem IV.5 to clarify the issue. For good qLDPC codes, the linear distance will imply a constant α and a non-zero β , thus our distance claim still holds without requirements on the energy barrier.

We acknowledge that linear energy barrier does not necessarily imply a constant β in small-set (co)boundary expansion, and we have adjusted the claim accordingly in Theorem II.1 and II.3. We would like to point out that the known constructions of good quantum LDPC codes have constant β , as shown in [5, 6].

For the reviewer’s second concern on good qLDPC having an expanding Tanner graph, we acknowledge that there is no formal proof of the theorem in the literature. We have adjusted the statement to “there is no geometrically local embedding as the parameter of good qLDPC codes violates the BPT bound.”.

For the reviewer’s third concern on balanced product codes, we acknowledge that the naming “balanced product codes” will cause confusion. We have changed the statement to LDPC codes that have a local product structure, which is the requirement in [1].

The other typos are fixed.

Manuscript revisions:

1. **page 2, line 109-118, Theorem II.1:** Given any family of quantum LDPC codes with asymptotically linear dimension, linear distance, there exists a family of D -dimensional geometrically local quantum codes with code dimension $k = \tilde{\Omega}(n^{\frac{D-2}{D}})$, code distance $d = \tilde{\Omega}(n^{\frac{D-1}{D}})$. Moreover, if the underlying complex of code family has constant (small-set) (co)boundary expansion at level 1, the new code family has energy barrier $\mathcal{E} = \tilde{\Omega}(n^{\frac{D-2}{D}})$, where we omit polylog(n) factors in the notation $\tilde{\Omega}$.
2. **page 4, line 293-298 Theorem II.3:** Given a good qLDPC code Q with parameter $[[n, k = \Theta(n), d = \Theta(n)]]$, the new code Q_L has parameter $[[n_L = \Theta(nL^2), k_L = k, d_L = \Theta(dL)]]$. Moreover, if X has constant (small-set) (co)boundary expansion at level 1, the new code Q_L has energy barrier $\mathcal{E}_L = \Theta(\mathcal{E}) = \Theta(n)$.
3. **page 7 line 534-535, Theorem IV.5, added:** Moreover, α is only determined by α_{qLDPC} .
4. **page 3, line 206-209:** It is known that the Tanner graph of a good quantum LDPC code Q cannot have a geometrically local embedding, since the code’s distance violates the bound in [9].
5. **page 2, line 76-80:** we showed that the step from code to manifold in Portnoy’s work can be avoided if we start from quantum LDPC codes that admit a local product structure [1,3].

Reviewer #2

This is a nicely presented work that provides an incremental extension of previous work by some of the same authors, following an earlier work by E. Portnoy. The main result is a mapping from arbitrary good qLDPC codes to local codes on finite dimensional lattices with (almost) the best possible code parameters that are achievable for such codes. I have included the qualifier almost because the main result of saturating optimal parameter scaling for local codes in any dimension, rather than achieving

this only up to 1/polylog factors, appears to depend on an unpublished work cited as Ref.16. There was no clear explanation of how this improved result is obtained beyond this citation. The main innovation of this work seems to lie in generalizing the family of codes that the construction applies to. As the authors point out, this offers promise of potential future applications including weight reduction of quantum codes. I tend to agree with the authors that this work is likely to lead to interesting follow ups, but I do not think it crosses the line for acceptance to Nature Communications in its current form given that it does not contain proof that it does indeed improve on the code properties obtained in previous works.

Author's Reply: We thank the reviewer for pointing out the problem of unpublished work [3]. After private communications with the authors, we learned that the manuscript is still under preparation and cannot be published online in the near future. We have changed the embedding theorem using the result of [3] to the theorem shown in [4] with extra polylog loss. Due to the adjustment in the embedding theorem, our main theorem will introduce a polylog loss in the asymptotic rate and distance, and we have changed our claim from optimal to almost optimal accordingly.

We would like to point out that our construction can be improved to optimal via better embedding results. We still firmly believe that our results develop substantial improvement over previous papers. Our results broaden the generality of construction to any quantum LDPC codes and can be embedded in any dimensions, and we also avoided the code to manifold construction due to our subdivision procedure.

Manuscript revisions: All of the optimal claims have been switched to almost optimal, and we included the polylog loss in our theorem statements.

Reviewer #3

The authors present a generalization of their earlier work [14], proposing a method to construct optimal geometrically local quantum codes from arbitrary good quantum LDPC codes. Compared to prior constructions, this approach offers improved generality as it applies to arbitrary spatial dimensions and is not limited to specific code families.

A particularly interesting aspect of the work is the possibility that one could convert a quantum LDPC code into another qLDPC code with similar properties but potentially reduced check weight, which could provide practical advantages. However, the extent of the improvement remains unclear, as the manuscript does not currently include concrete examples or performance evaluations to support or illustrate the practical advantages of the construction.

There are two points that could be strengthened to evaluate the significance to the field:

- 1) The paper does not provide explicit examples of geometrically local quantum codes generated using the proposed method. Including such examples with numerical simulations, demonstrating decoding performance, would allow for comparisons with existing solutions such as bivariate bicycle codes or surface codes. Metrics such as overhead in terms of physical qubits and logical error rate under standard noise models would be particularly useful.
- 2) While the paper emphasizes geometric locality, it would be valuable to discuss how the construction might apply in non-local architectures, such as those incorporating c-couplers or photonic interconnects that support long-range qubit interactions. Clarifying whether the method still offers advantages in such settings, or how it could be adapted, would broaden its relevance.

Author's Reply: We thank the reviewer for raising questions on numerical simulation and experimental long-range actions. We acknowledge that these points are important for practical applications. However, the major focus of our paper is on theoretical constructions of optimal geometrically local codes. As the constructions of good quantum LDPC codes [7, 8, 5] and other optimal geometrically local codes [4, 9] only focus on the asymptotic behaviors of the code, we choose to follow the convention and not include numerical simulation in our current paper.

References

- [1] Ting-Chun Lin, Adam Wills, and Min-Hsiu Hsieh. *Geometrically Local Quantum and Classical Codes from Subdivision*. Sept. 2023.
- [2] Xingjian Li, Ting-Chun Lin, and Min-Hsiu Hsieh. *Transform Arbitrary Good Quantum LDPC Codes into Good Geometrically Local Codes in Any Dimension*. Aug. 2024.
- [3] Ting-Chun Lin and Elia Portnoy. *In preparation*. 2025.
- [4] Elia Portnoy. *Local Quantum Codes from Subdivided Manifolds*. June 2023.
- [5] Irit Dinur, Min-Hsiu Hsieh, Ting-Chun Lin, and Thomas Vidick. “Good Quantum LDPC Codes with Linear Time Decoders”. In: *Proceedings of the 55th Annual ACM Symposium on Theory of Computing*. STOC 2023. New York, NY, USA: Association for Computing Machinery, June 2023, pp. 905–918.
- [6] Anurag Anshu, Nikolas P. Breuckmann, and Chinmay Nirkhe. “NLTS Hamiltonians from Good Quantum Codes”. In: *Proceedings of the 55th Annual ACM Symposium on Theory of Computing*. STOC 2023. New York, NY, USA: Association for Computing Machinery, June 2023, pp. 1090–1096.
- [7] Pavel Panteleev and Gleb Kalachev. “Asymptotically Good Quantum and Locally Testable Classical LDPC Codes”. In: *Proceedings of the 54th Annual ACM SIGACT Symposium on Theory of Computing*. STOC 2022. New York, NY, USA: Association for Computing Machinery, June 2022, pp. 375–388.
- [8] Anthony Leverrier and Gilles Zémor. “Quantum Tanner Codes”. In: *2022 IEEE 63rd Annual Symposium on Foundations of Computer Science (FOCS)*. IEEE Computer Society, Oct. 2022, pp. 872–883.
- [9] Dominic J. Williamson and Nouédyn Baspin. *Layer Codes*. Sept. 2023.